# The Impact of Medication Regimen Adjustment Ratio on Adherence and Glycemic Control in Patients with Type 2 Diabetes and Mild Cognitive Impairment

**DOI:** 10.3390/biomedicines12092110

**Published:** 2024-09-16

**Authors:** Xiaoqing Song, Jingwei Wu, Yan Yang, Weijie Xu, Xiaoli Shi, Kun Dong, Mengni Li, Xi Chen, Yuping Wang, Xuna Bian, Lianlian Gao, Xuefeng Yu, Delin Ma, Jing Tao

**Affiliations:** 1Department of Endocrinology, Tongji Hospital, Tongji Medical College, Huazhong University of Science and Technology, Wuhan 430030, China; song2014xs@163.com (X.S.); 13908641637@163.com (J.W.);; 2Branch of National Clinical Research Center for Metabolic Diseases, Wuhan 430030, China; 3Department of Nursing, Tongji Hospital, Tongji Medical College, Huazhong University of Science and Technology, Wuhan 430030, China

**Keywords:** glycemic control, medication adherence, type 2 diabetes mellitus (T2DM), mild cognitive impairment (MCI), cognitive status

## Abstract

Background: An antidiabetic medication regimen is crucial for maintaining glycemic control. Type 2 diabetes mellitus (T2DM) and cognitive dysfunction have a bidirectional relationship. This study aims to explore the impact that adjusting antidiabetic medication regimens has on medication adherence, glycemic control, and cognitive function in patients with T2DM and mild cognitive impairment (MCI). Methods: This is an observational cross-sectional analysis that includes 364 consecutive inpatients with T2DM. Clinical data were collected, medication adherence was assessed using the Medication Adherence Report Scale (MARS-5), and cognitive status was evaluated using the Chinese version of the Montreal Cognitive Assessment (MoCA) and Mini-mental State Examination (MMSE). These data were obtained both during hospitalization and at a three-month follow-up. Multivariable logistic regression analysis was applied to determine the association between changes in medication regimens and medication adherence, glycemic control, and cognitive function. Results: Baseline medication adherence was high across all three different cognitive status groups, with no significant difference in MARS-5 scores. At the 3-month follow-up, the group with a high adjustment ratio of antidiabetic medication regimens showed an increase in their hemoglobin A1c (HbA1c) level compared to the baseline, while the group with a low adjustment ratio showed a decrease in this level. In addition, the MoCA, MMSE, and MARS-5 scores of the high-adjustment group were significantly lower than those of the low-adjustment group. Conclusions: A high ratio of medication adjustment was significantly associated with worse medication adherence and glycemic control in T2DM patients with MCI. Patients with a low ratio of medication adjustment had good adherence and better glycemic control. Clinicians should take cognitive status into account when adjusting antidiabetic regimens for T2DM patients and may need to provide additional guidance to patients with cognitive impairment to improve adherence and glycemic outcomes.

## 1. Introduction

The global prevalence of diabetes mellitus is rapidly increasing, with an estimated 783 million people expected to be affected worldwide by 2045 [1]. The primary therapeutic goal for individuals with diabetes is to achieve and maintain glycemic control and prevent diabetes-related complications, morbidity, and mortality [2]. Proper management through antidiabetic medications and recommended lifestyle changes is crucial for maintaining glycemic control and preventing diabetes-related complications [3]. As diabetes progresses, many patients often require ongoing adjustments to their medication regimens to achieve or maintain glycemic control, making medication adherence a critical aspect of diabetes management [4].

Cognitive dysfunctions, including mild cognitive impairment (MCI) and dementia (including conditions such as Alzheimer’s disease), are gaining increasing recognition as common complications of Type 2 diabetes mellitus (T2DM) [5,6,7]. Glycemic control in T2DM patients is closely related to cognitive function changes. Poor glycemic control is associated with cognitive decline [8,9], while cognitive impairment can make self-care and glycemic control more challenging [10]. Thus, stable glucose control is essential for T2DM patients with mild cognitive impairment.

Poor medication adherence can result in inadequate glycemic control, increased use of medical resources, higher medical costs, and significantly higher mortality [11,12,13]. Previous research has identified several factors that influence long-term medication adherence, including patient factors (such as age, education level, and depression), environmental factors (such as social support and socioeconomic status), and treatment regimen factors (such as drug burden, regimen complexity, side effects, required treatment duration, and administration time) [14,15]. Recent studies have primarily focused on the impact of medication regimen complexity on adherence and glycemic control [14,16,17]. Additionally, there is limited research on medication adherence in diabetes patients with cognitive impairment. Some studies have found no association between dementia and adherence to antidiabetic medications [18], while others have highlighted how negative beliefs about medications and regimen-related distress affect adherence and glycemic control in patients with diabetes and mild cognitive impairment (MCI) [19]. Currently, there is a lack of research on the impact of medication regimen adjustments on adherence, glycemic control, and cognitive status in T2DM patients with cognitive impairment. Understanding the relationship between medication regimen adjustments and adherence and glycemic control may help with the development of future interventions to improve treatment outcomes. This study aims to explore the impact of changes in antidiabetic medication regimens on medication adherence, glycemic control, and cognitive status in T2DM patients with MCI.

## 2. Materials and Methods

### 2.1. Study Design and Participants

A total of 521 patients with T2DM who were hospitalized at Tongji Hospital (Wuhan, China) between September 2023 and December 2023 were enrolled in this study. And 157 patients were excluded based on the exclusion criteria, leaving 364 patients included in this study. Baseline interviews assessed the patients’ cognition status and medication adherence to antidiabetic drugs upon admission, and follow-up interviews were conducted 3 months post-discharge. Exclusion criteria included (1) other types of diabetes; (2) age under 18 years; (3) acute cardiovascular and cerebrovascular events (such as cerebral infarction); (4) metabolic diseases that may have affected cognitive function in the last 3 months, such as severe hypoglycemia, diabetic ketoacidosis, diabetic hyperglycemia hypertonic coma, hypothyroidism, etc.; (5) combined with head trauma that might affect cognitive function; mental and neurological disorders such as depression, anxiety, delirium; severe lung or kidney diseases, history of heart failure, malignant tumors, etc.; (6) missing 3-month follow-up data.

A total of 194 T2DM patients with mild cognitive impairment were followed up for data on medication adherence, glycemic control, and cognitive status three months after their initial hospitalization.

The study was approved by the Ethics Committee of Tongji Hospital and was conducted according to the Declaration of Helsinki. Appropriate consent and assent were acquired from all participants.

### 2.2. Data Collection Procedure and Methods

Clinical and demographic data and diabetes-related information were obtained from Tongji Hospital’s hospitalization medical record system. The collected data included the patient’s sex, age, weight, height, education level, diabetes duration, any hypoglycemic events they had experienced in the last 3 months, any history of hyperlipidemia or hypertension, any history of smoking and drinking, coronary heart disease, or cerebrovascular disease, and details of antidiabetic medication regimens (including before hospitalization and at discharge, and the drugs types and dosages prescribed).

### 2.3. Glycemic Control

Hemoglobin A1c (HbA1c) values were retrieved from the patients’ medical records to determine the levels at the baseline interview and at the 3-month follow-up. Higher HbA1c levels indicate worse glycemic control.

### 2.4. Medication Adherence

Medication adherence was estimated by the Medication Adherence Report Scale (MARS-5). The MARS-5 was developed by Horne et al. [20] and has been widely used in studies on various chronic illnesses, including T2DM, hypertension, and asthma [21,22,23]. The MARS-5 has good reliability and validity and might be the most accurate self-report [24,25]. The score ranges from 5 to 25, and a higher MARS-5 score indicates higher self-reported adherence. In the study, a cut-off point of 90% [26] was used and the patients with total scores ≥ 23 were considered adherent, while those with scores of less than 23 were considered non-adherent [26].

### 2.5. Cognition Status

Cognition was assessed using the Chinese versions of the Montreal Cognitive Assessment (MoCA) and Mini-mental State Examination (MMSE), which are both valid and reliable in Chinese populations [27,28,29,30]. The scores of the MMSE and MoCA, respectively, range from 0 to 30; higher scores indicate better cognition. The MMSE score was bounded by 23/24 for dementia [31] and the MOCA score was bounded by 25/26 for MCI; 1 point was added to the total MoCA score for participants with 12 years of education or fewer [32,33].

### 2.6. Measures

Medication adherence and cognition status were assessed through interviews at admission and at a 3-month follow-up assessment.

#### 2.6.1. The MoCA and MMSE Assessments Were Conducted as Follows

(1) The assessments took place in a dedicated, quiet room with necessary materials provided, and no clocks or calendars present.

(2) Each participant took the test face to face.

(3) A 5 min calming conversation preceded the assessment to help the participant relax.

(4) Each test item was attempted only once, with neutral feedback.

(5) The assessment lasted about 10 min with uniform instructions and adherence to “Scoring Criteria”.

#### 2.6.2. MARS-5 Assessment

(1) Medication adherence was assessed based on the responses to five questions (e.g., “I forget to take my antidiabetic drugs”; “I alter the dose of my antidiabetic drugs”; “I stop taking my antidiabetic drugs for a while/sometimes”; “I decide to skip one of my antidiabetic drugs dosages”; “I use my antidiabetic drugs less than is prescribed”, using a 5-level response format (1—always, 2—often, 3—sometimes, 4—rarely, and 5—never).

(2) Scores ranged from 5 to 25, with higher scores indicating better adherence.

All assessments were performed by clinicians who are trained to carry out these tasks.

### 2.7. Adjustment Ratio of Medication

Adjustments in the antidiabetic medication regimen at discharge were quantified using the medication adjustment ratio, which was calculated as follows: 

Adjustment ratio = (Number of Newly Added Antidiabetic Drug Types)/(Total Number of the Antidiabetic Drug Types at discharge) × 100%.

Ratios were categorized as low (0–33.3%), moderate (33.4–66.6%), and high (66.7–100%).

The adjustment of antidiabetic medications for all inpatients was a comprehensive evaluation performed by attending physicians with over 10 years of clinical experience, based on the latest diabetes treatment guidelines [34,35,36], clinical experience, and the patient’s blood glucose and HbA1c levels during hospitalization.

### 2.8. Statistical Analyses

Data are presented as the median with the interquartile range for non-normally distributed continuous variables or means with standard deviations (SD) for normally distributed continuous variables and as counts (percentages) for categorical variables. The modified Kolmogorov–Smirnov test was used to test the normal distribution of continuous variables. The Mann–Whitney U-test was used for the two-group comparison for non-normally distributed variables. The χ^2^ test or Fisher’s exact test was used to determine the categorical variables. The Kruskal–Wallis test was used to evaluate the differences between groups with different cognitive statuses or different ratios of antidiabetic medication adjustment. To identify the independent associated variable for non-adherence or worse glycemic control, we used a univariate logistic regression model to identify multiple variables. Then, we applied a multivariate stepwise logistic regression model to exclude confounding factors and identify independent factors that were significantly associated with medication non-adherence or poor glycemic control. After collinearity was excluded, the variables selected for inclusion in the multivariate stepwise model were sex, age, low education, HbA1c%, MoCA, insulin use, and the adjustment ratio of the medication regimen. Statistical analyses were performed using SPSS 26.0 software (IBM Corp, Armonk, NY, USA). *p* < 0.05 was considered statistically significant.

## 3. Results

### 3.1. Characteristics of Study Population

The flow chart of our study is shown in Figure 1. Among the 521 inpatients diagnosed with T2DM, 157 patients were excluded according to the exclusion criteria and 364 patients were included in the study. The included participants were categorized into three groups based on their cognition status: normal cognition (78, 21.4%), mild cognitive impairment (194, 53.3%), or dementia (92, 25.3%). The baseline characteristics of the three groups are summarized in Table 1. Significant differences were observed among the groups concerning the patients’ age, gender, diabetes duration, hyperlipidemia, hypertension, education level, and HbA1c (all *p* < 0.05). Patients with worse cognitive status were older and had a longer duration of diabetes, lower education levels, and higher rates of hyperlipidemia and hypertension. No significant differences were found for BMI, hypoglycemic events, smoking, or drinking status (all *p* > 0.05). The MARS-5 scores were high across all groups (93.6% for normal, 96.9% for MCI, and 94.6% for dementia) with no significant difference (*p* = 0.412), indicating good medication adherence in all groups.

### 3.2. Characteristics of MCI Group after MedicationAdjustment

Since some MCI patients might revert to normal cognitive function, we examined changes in HbA1c, MoCA, MMSE, and MARS-5 scores at a 3-month follow-up post-medication change in T2DM patients with MCI. To assess the association between these factors and medication adjustment, MCI patients were divided into three groups based on the medication adjustment ratio at discharge (Table 2). No significant differences in age, sex, diabetes duration, hyperlipidemia, hypertension, history of smoking and drinking, and diabetes complications were observed among the groups (*p* > 0.05). Significant differences in baseline HbA1c levels were noted among the groups (*p* < 0.001), with higher adjustment ratios corresponding to higher baseline HbA1c levels. No significant differences were found in MoCA scores (*p* = 0.794), MMSE scores (*p* = 0.421), or education levels (*p* = 0.554). MARS-5 scores were similarly high across groups (96.0%, 98.0%, and 96.8%) with no significant difference (*p* = 0.844). At the 3-month follow-up, significant changes were observed in HbA1c, MoCA, MMSE, and MARS-5 scores across the groups. The high-ratio group exhibited a higher increase (0.4 (−0.6,0.6)) in HbA1c than the moderate-ratio group (0.1 (−0.5,0.5)), while the low-ratio group exhibited a decrease (−0.2 (−0.7,0.2)) in HbA1c. The MoCA, MMSE, and MARS-5 scores were significantly lower in the high-ratio group compared to the other two groups (all *p* < 0.05). To control for medication regimen differences, MCI patients were further subdivided based on glucose-lowering medication regimens. Analyses of subgroup1 and subgroup2 revealed similar changing trends in HbA1c, MoCA, and MARS-5 scores (Table 3). Analyses of subgroups 3 and 4 revealed no significant differences in cognitive or adherence scores (Appendix A).

### 3.3. Factors Influencing Medication Adherence

Logistic regression analysis was performed to identify factors associated with non-adherence in T2DM patients with MCI following medication changes. The baseline variables of MoCA score, Hb1Ac, level of education, insulin use, and adjustment ratio were included in the logistic regression, as these variables differed significantly among patients with different medication adherence (Figure 2). The MoCA score, Hb1Ac level, low education, insulin use, and medication adjustment ratio were related to the risk of non-adherence in T2DM patients with MCI in the univariate analysis (*p* < 0.05 for each) (Table 4), while multivariate analysis confirmed that the MoCA score (OR: 0.73, 95% CI: 0.62–0.87, *p* < 0.001) and medication adjustment ratio were independently associated with medication adherence after adjusting for age, sex, education, insulin use, and Hb1Ac level. There was a nearly 3-fold and 11-fold increased risk of non-adherence in patients with adjustment ratios of 33.4–66.6% (OR: 3.22, 95% CI: 1.01–10.24, *p* = 0.048) and adjustment ratios of 66.7–100% (OR: 10.90, 95% CI: 3.63–32.76, *p* < 0.001), respectively, compared with patients with adjustment ratios of 0–33.3%.

### 3.4. Factors Influencing Glycemic Control

Logistic regression analysis was conducted to identify factors affecting glycemic control in T2DM patients with MCI following medication changes. Patients were categorized into two groups based on ΔHbA1c values, using 0 as the cutoff value (Table 5). Univariate analysis indicated that higher HbA1c (OR: 0.87, 95% CI 0.76–0.99, *p* = 0.039) at baseline and a higher medication adjustment ratio were associated with worse glycemic control. Multivariate analysis demonstrated that HbA1c at baseline and the medication adjustment ratio remained significant predicators after adjusting for age, sex, and MoCA. Patients with adjustment ratios of 33.4–66.6% (OR: 2.70, 95% CI: 1.13–6.41, *p* = 0.025) and adjustment ratios of 66.7–100% (OR: 4.57, 95% CI: 2.01–10.40, *p* < 0.001) had nearly 3-fold and 4.5-fold increased risk of worse glycemic control, respectively, compared with patients with adjustment ratios of 0–33.3% after multiple adjustments.

## 4. Discussion

The current study focused on the impact of medication adjustments on glycemic control, cognitive status, and medication adherence in patients with Type 2 diabetes mellitus and mild cognitive impairment. The results indicated that a high ratio of adjustments to antidiabetic medications was associated with worse medication adherence, poorer glycemic control, and cognitive decline in patients with T2DM and mild cognitive impairment. Patients with a low ratio of medication adjustment had good adherence and better glycemic control.

More than half of the enrolled T2DM patients had MCI, and nearly a quarter had dementia, which was similar to our previous findings [37]. Consistent with previous studies, diabetic patients with more severe cognitive impairment were older, had a higher probability of hyperlipidemia and hypertension, and had a longer disease duration [38,39]. Consistent with previous studies, we also noticed that despite cognitive impairment, medication adherence remained high [19,40], possibly due to medication habits or support from family members.

Our finding suggested that higher medication adjustment ratios were associated with poorer medication adherence, worse glycemic control, and worse cognitive status in T2DM patients with MCI. The significant association between medication adjustment ratio and glycemic control observed in our study is a relatively novel finding. T2DM patients with high medication regimen complexity had poor medication adherence and this was associated with poor glycemic control [14]. The increased risk of worse glycemic control and worse cognitive status with higher medication adjustment ratios suggests that a high ratio of adjustments to the medication regimen may disrupt the stability of diabetes management, particularly in patients with MCI, highlighting the importance of cautious medication management and personalized treatment plans that consider the patient’s cognitive function.

Subgroup analyses were in line with previous findings [41] that patients receiving insulin were at higher risk of non-adherence to medication following a high ratio of medication regimen adjustments. This might be due to the fact that insulin regimens are generally more complex, requiring greater management skills, as well as greater family/social support. The need for insulin injections and the technical skill required for insulin administration may pose challenges for patients, particularly those with cognitive impairment. Simplifying insulin regimens and providing ongoing support might be necessary to improve adherence in patients with cognitive dysfunction.

Additionally, our results also suggested that the adjustment ratio of medication was a significant risk factor for non-adherence, a finding that is consistent with a recently published study about medication regimen complexity [14]. In diabetes treatment, in addition to factors related to the treatment regimen, many other factors affect medication adherence. This study showed that age, education level, baseline HbA1c levels, and insulin use were also identified as risk factors for adherence, while the MoCA score was a protective factor for adherence.

Changes in medication regimens also pose a risk for poor glycemic control. The association between an increased ratio of diabetes regimen adjustment and poor glycemic control suggests that a high ratio of modifications to glycemic regimens may exacerbate the disease burden in patients with diabetes without improving their glycemic outcomes. Similar previous studies support the finding that high diabetes-specific medication regimen complexity for antidiabetic agents is associated with poorer glycemic control, which is possibly related to decreased adherence [14,18].

Changes in medication regimens may lead to increased confusion or difficulty adhering to prescribed treatments for patients with T2DM and MCI. Additionally, poor medication adherence itself leads to worse glycemic control. Previous studies have reported that long-term diabetes and poor blood glucose control (including hyperglycemia, hypoglycemia, and glycemic fluctuations) can lead to neuronal damage and inflammation-related glial activation by disrupting the blood–brain barrier and altering the brain’s metabolism, resulting in progressive neuropathy and ultimately leading to cognitive impairment [42,43].

Our results highlight that considering cognitive impairment in T2DM patients is crucial for effective management. Cognitive impairment, such as MCI, can affect a patient’s ability to follow complex medication regimens. Therefore, clinicians should take into account the cognitive status of patients when making adjustments to antidiabetic medications. Meanwhile, the MARS-5 we used has been proven to have good validity and reliability in patients with diabetes [24,25]. However, our study also had several limitations. First, MARS-5 scores were used to assess medication adherence, due to the lack of more objective assessment methods to confirm medication adherence. Second, the relatively short follow-up period may have limited our ability to detect substantial changes in medication adherence, glycemic control, and MoCA scores between enrollment and follow-up. Third, some potential confounding variables that might influence medication adherence or glycemic control, such as socioeconomic status, other comorbidities, and their related medications, were not available for analysis. Finally, this is a single-center study with a single source of patients, and the patient’s clinical characteristics and hospital environment may not be representative of the broader population of T2DM patients with mild cognitive impairment. Admittedly, a 3-month follow-up with a single center may not reflect long-term glycemic control and cognitive function, but studies on how medication adjustments affect glycemic control, cognitive status, and medication adherence are lacking. Future research will require prospective multicenter studies with larger sample sizes to better investigate the long-term effects of adjusting antidiabetic medication regimens on diabetes management and cognitive function in patients with T2DM and mild cognitive impairment.

## 5. Conclusions

In summary, our study is the first to reveal a significant association between a high ratio of adjustment of the antidiabetic medication regimens and poor medication adherence, worse glycemic control, and decreased cognitive status in T2DM patients with mild cognitive impairment. This study offers valuable insights into the cognitive challenges faced by T2DM patients with mild cognitive impairment when it comes to managing their diabetes. For patients with a high ratio of adjustment in their medication regimen, more education and support are needed to improve medication adherence and glycemic control and to facilitate the recovery of cognitive function or delay cognitive decline.

## Figures and Tables

**Figure 1 biomedicines-12-02110-f001:**
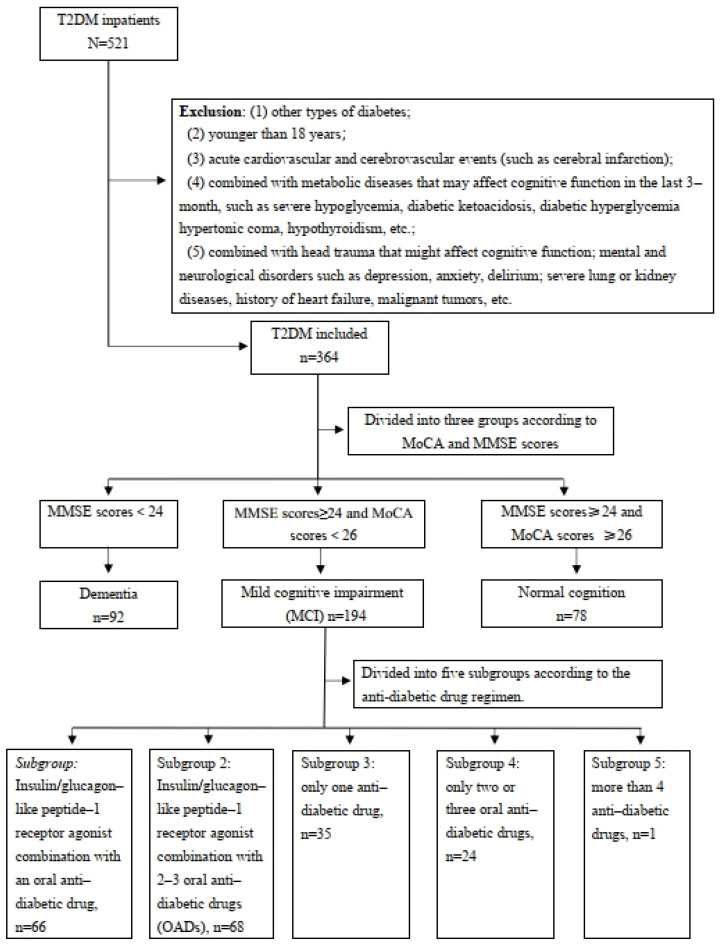
Flow chart of the study population.

**Figure 2 biomedicines-12-02110-f002:**
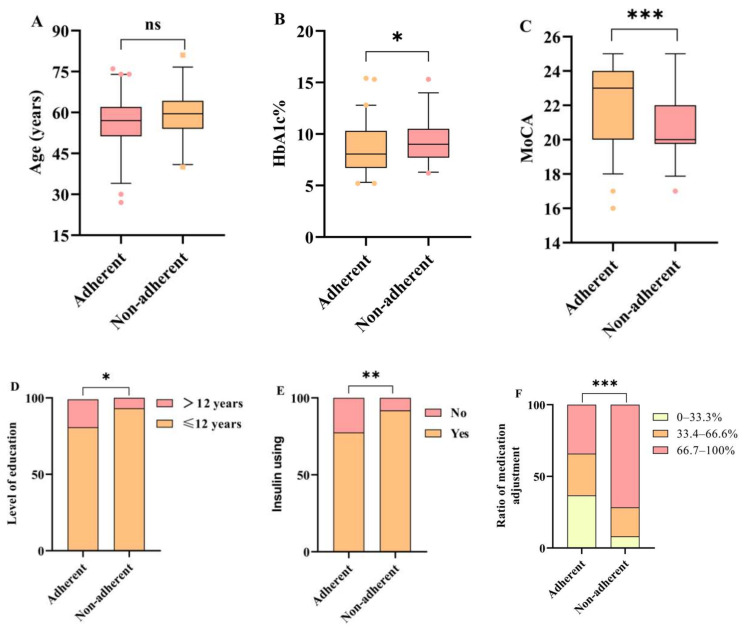
Comparisons of baseline variables in two groups classified by level of medication adherence in T2DM patients with MCI. (**A**–**F**) Age, HbA1c%, MoCA, level of education, insulin using and ratio of medication adjustment in the two groups were assessed. Adherent—MARS-5 scores ≥ 23; Non-adherent—MARS-5 scores < 23. *, *p* < 0.05; **, *p* < 0.01; ***, *p* < 0.001; ns, not significant.

**Table 1 biomedicines-12-02110-t001:** Baseline demographic, clinical, and laboratory characteristics of the type 2 diabetic patients categorized by cognition status (*n* = 364).

Characteristics	Normal (*n* = 78)	MCI (*n* = 194)	Dementia (*n* = 92)	*p*
Age (years)	42.0 (33.0, 50.3)	58.0 (52.0, 62.0)	66.0 (59.0, 72.0)	<0.001
Gender				
Male, *n* (%)	64 (82.1)	127 (65.5)	39 (42.4)	<0.001
Female, *n* (%)	14 (17.9)	67 (34.5)	53 (57.6)	
Diabetes duration (years)	3 (0.0, 6.0)	8 (3.0, 14.0)	12 (6.0, 16.8)	<0.001
BMI (kg/m^2^)	23.4 (22.0, 24.5)	23.3 (22.2, 25.0)	23.5 (22.8, 24.6)	0.688
Hypoglycemic event, *n* (%)	3 (3.8)	8 (4.1)	6 (6.5)	0.620
Hyperlipidemia, *n* (%)	56 (71.8)	138 (71.1)	84 (91.3)	<0.001
Hypertension, *n* (%)	25 (32.1)	92 (47.4)	74 (80.4)	<0.001
Smoker, *n* (%)	10 (12.8)	26 (13.4)	13 (14.1)	0.969
Drinker, *n* (%)	7 (8.9)	22 (11.3)	8 (8.7)	0.729
Low education (≤12 years), *n* (%)	34 (43.6)	166 (85.6)	90 (97.8)	<0.001
MARS-5 ≥ 23	73 (93.6)	188 (96.9)	87 (94.6)	0.412
<23	5 (6.4)	6 (3.1)	5 (5.4)	
HbA1c (%)	9.6 (8.0, 10.8)	8.4 (7.1, 10.3)	9.3 (7.5, 10.8)	0.035
MoCA	28.0 (27.0, 30.0)	22.0 (20.0, 24.0)	15.0 (12.0, 18.0)	<0.001
MMSE	30.0 (30.0, 30.0)	26.5 (25.0, 28.0)	20.0 (18.0, 22.0)	<0.001

Data are expressed as median (quartile) or *n* (%). MCI—mild cognitive impairment; BMI—body mass index; MARS-5—the Medication Adherence Report Scale-5; HbA1c—hemoglobin A1c; MoCA—Montreal Cognitive Assessment; MMSE—Mini-mental State Examination.

**Table 2 biomedicines-12-02110-t002:** Characteristics of the MCI group categorized by adjustment ratio of antidiabetic drugs (*n* = 194).

Characteristics	Adjustment Ratio of Antidiabetic Drugs			*p*
	0–33.3% (*n* = 50)	33.4–66.6% (*n* = 50)	66.7–100% (*n* = 94)	
Baseline				
Age (years)	58.0 (52.0, 62.0)	59.5 (52, 62.3)	58.0 (52.0, 62.0)	0.866
Gender				
Male, *n* (%)	30 (60)	38 (76)	60 (63.8)	0.199
Female, *n* (%)	20 (40)	12 (24)	34 (36.2)	
Diabetes duration (years)	10.0 (4.0, 13.3)	8.0 (5.0, 15.3)	23.8 (21.9, 25.3)	0.181
BMI (kg/m^2^)	23.3 (22.3, 24.9)	23.8 (21.9, 25.3)	23.2 (22.2, 25.0)	0.934
Hypoglycemic event, *n* (%)	3 (6.0)	3 (6.0)	2 (2.1)	0.401
Hyperlipidemia, *n* (%)	32()	33()	73()	0.169
Hypertension, *n* (%)	22 (44.0)	24 (48.0)	46 (48.9)	0.849
Smoker, *n* (%)	7 (14.0)	9 (18.0)	10 (10.6)	0.462
Drinker, *n* (%)	6 (12.2)	7 (14.0)	9 (9.7)	0.727
Diabetes Complications, *n* (%)				
Diabetic Foot	2 (4.0)	3 (6.0)	2 (2.1)	0.488
Diabetic Retinopathy	11 (22.0)	15 (30.0)	30 (31.9)	0.448
Diabetic Nephropathies	16 (32.0)	17 (34.0)	38 (40.4)	0.550
Diabetic Peripheral Neuropathies	32 (64.0)	31 (62.0)	64 (68.1)	0.741
Diabetic Peripheral vascular disease	38 (76)	37 (74)	79 (84)	0.289
HbA1c %	7.1 (6.1, 8.5)	8.4 (7.2, 10.3)	9.2 (7.9, 10.7)	<0.001
MoCA	22.0 (20.0, 24.0)	22.0 (20.0, 24.0)	22.0 (20.0, 24.0)	0.794
MMSE	27.5 (25.8, 29.0)	27.0 (25.0, 28.0)	26.0 (25.0, 28.0)	0.421
Low education	41 (82.0)	42 (84.0)	83 (88.3)	0.554
MARS-5, ≥23	48 (96.0)	49 (98.0)	91 (96.8)	0.844
<23	2 (4.0)	1 (2.0)	3 (3.2)	
3-month follow-up				
ΔHbA1c%	−0.2 (−0.7, 0.2)	0.1 (−0.5, 0.5)	0.4 (−0.6, 0.6)	0.003
ΔMoCA	1.0 (0, 2.0)	1.5 (−1.0, 2.0)	−1.0 (−2.0, 2.0)	<0.001
ΔMMSE	0.0 (0, 0)	0.0 (0, 0)	0.0 (−1.0, 0)	<0.001
MARS-5, ≥23	44 (88.0)	35 (70.0)	41 (43.6)	<0.001
<23	6 (12.0)	15 (30.0)	53 (56.4)	

Data are expressed as median (quartile) or *n* (%). MCI—mild cognitive impairment; HbA1c—hemoglobin A1c; MoCA—Montreal Cognitive Assessment; MMSE—Mini-mental State Examination; MARS-5—the Medication Adherence Report Scale-5; ΔHbA1c—HbA1c_3-month_−HbA1c_baseline_; ΔMoCA—MoCA_3-month_−MoCA_baseline_; ΔMMSE—MMSE_3-month_−MMSE_baseline_.

**Table 3 biomedicines-12-02110-t003:** Subgroup analysis of the MCI patients categorized by antidiabetic medication regimen.

Subgroup	Characteristics	Adjustment Ratio of Antidiabetic Drugs			*p*
		0–33.3%	33.4–66.6%	66.7–100%	
1	Baseline				
	*n* (%)	11 (16.7)	23 (34.8)	32 (48.5)	
	Age (years)	60.0 (57.0, 62.0)	60.0 (55.0, 65.0)	60.0 (52.0, 64.8)	0.917
	Diabetes duration (years)	10.0 (5.0, 13.0)	14.0 (5.0, 20.0)	7.0 (2.0, 16.5)	0.383
	HbA1c%	7.8 (6.9, 8.4)	8.1 (7.4, 10.1)	9.0 (7.8, 11.4)	0.051
	MoCA	22.0 (20.0, 24.0)	22.0 (19.0, 24.0)	21.0 (20.0, 22.8)	0.932
	MMSE	26.0 (25.0, 28.0)	26.0 (25.0, 28.0)	26 (25.0, 28.0)	0.906
	Low education	10 (90.9)	18 (78.3)	28 (87.5)	0.531
	MARS-5, ≥23	10 (81.8)	22 (100.0)	32 (100.0)	0.285
	<23	1 (18.2)	1	0	
	3-month follow-up				
	ΔHbA1c%	−0.2 (−0.9, 0.0)	0.1 (−0.5, 0.5)	0.6 (−0.8, 0.8)	0.037
	ΔMoCA	2.0 (1.0, 2.0)	0.0 (−1.0, 2.0)	−2.0 (−2.0, −0.2)	<0.001
	ΔMMSE	0.0 (0.0, 1.0)	0.0 (0.0, 0.0)	0.0 (−1.0, 0.0)	<0.001
	MARS-5, ≥23	11 (100.0)	12 (52.2)	8 (25.0)	<0.001
	<23	0 (0)	11 (47.8)	24 (75.0)	
2	Baseline				
	*n* (%)	13 (19.1)	17 (25.0)	38 (55.9)	
	Age (years)	59.0 (51.5, 62.5)	60.0(52.0, 62.5)	55.5 (50.0, 61.0)	0.627
	Diabetes duration (years)	13.0 (2.5, 21.0)	6.0 (5.0, 10.0)	5.5 (1.8, 10)	0.187
	HbA1c%	7.3 (6.7, 8.8)	9.0 (8.0, 10.8)	10.2 (9.1, 10.7)	0.001
	MoCA	23.0 (20.0, 25.0)	23.0 (21.0, 23.5)	22.5 (20.0, 25.0)	0.785
	MMSE	28.0 (25.0, 29.0)	28.0 (25.52, 9.5)	27.5 (25.0, 29.0)	0.827
	Low education	11 (86.4)	16 (94.1)	33 (86.8)	0.670
	MARS-5, ≥23	13 (100.0)	17 (100.0)	36 (94.7)	0.443
	<23	0	0	2 (5.3)	
	3-month follow-up				
	ΔHbA1c%	−0.7 (−1.3, −0.2)	−0.1 (−0.5, 0.3)	0.4 (−0.6, 0.6)	0.009
	ΔMoCA	1.0 (0.0, 1.5)	2.0 (1.0, 2.0)	−1.0 (−2.0, 1.0)	0.001
	ΔMMSE	0.0 (0.0, 0.0)	0.0 (0.0, 0.0)	0.0 (−1.0, 0.0)	0.013
	MARS-5, ≥23	12 (92.3)	14 (82.4)	17 (44.7)	0.002
	<23	1 (7.7)	3 (17.6)	21 (55.3)	

Data are expressed as median (quartile) or *n* (%). MCI—mild cognitive impairment; Subgroup1—insulin/glucagon-like peptide-1 receptor agonists combined with an oral antidiabetic drug; subgroup2, insulin/glucagon-like peptide-1 receptor agonists combined with two or three oral antidiabetic drugs; HbA1c—hemoglobin A1c; MoCA—Montreal Cognitive Assessment; MMSE—Mini-mental State Examination; MARS-5—the Medication Adherence Report Scale-5; ΔHbA1c—HbA1c_3-month_−HbA1c_baseline_; ΔMoCA—MoCA_3-month_−MoCA_baseline_; ΔMMSE—MMSE_3-month_−MMSE_baseline_.

**Table 4 biomedicines-12-02110-t004:** Logistic regression analysis of factors associated with medication non-adherence.

Variables	Univariable		Multivariate	
	OR (95% CI)	*p*	OR (95% CI)	*p*
Male	0.69 (0.38–1.27)	0.234	0.81 (0.39–1.70)	0.581
Age	1.04 (1.01–1.07)	0.025	1.03 (0.99–1.08)	0.120
Low education	3.27 (1.19–9.03)	0.022	2.13 (0.67–6.83)	0.202
HbA_1_c%	1.18 (1.03–1.36)	0.018	1.04 (0.86–1.25)	0.710
MoCA	0.72 (0.62–0.83)	<0.001	0.73 (0.62–0.87)	<0.001
Insulin using	3.29 (1.29–8.41)	0.013	1.51 (0.49–4.68)	0.479
Adjustment ratio				
0–33.3%	ref		Ref	
33.4–66.6%	3.14 (1.10–8.94)	0.032	3.22 (1.01–10.24)	0.048
66.7–100%	9.48 (3.68–24.39)	<0.001	10.90 (3.63–32.76)	<0.001

All variables are used the baseline data. HbA1c—hemoglobin A1c; MoCA—Montreal Cognitive Assessment.

**Table 5 biomedicines-12-02110-t005:** Logistic regression analysis of factors associated with worse glycemic control.

Variables	Univariable		Multivariate	
	OR (95% CI)	*p*	OR (95% CI)	*p*
Male	1.14 (0.63–2.07)	0.669	1.32(0.69–2.51)	0.408
Age	1.03 (1.00–1.06)	0.083	1.02 (0.99–1.06)	0.250
Low education	1.51 (0.67–3.37)	0.318		
HbA_1_c%	0.87 (0.76–0.99)	0.039	0.77 (0.65–0.90)	0.002
MoCA	0.92 (0.81–1.05)	0.226	0.92 (0.79–1.06)	0.262
Insulin using	0.89 (0.42–1.89)	0.759		
Adjustment ratio				
0–33.3%	ref		ref	
33.4–66.6%	1.91 (0.86–4.22)	0.111	2.70 (1.13–6.41)	0.025
66.7–100%	2.53 (1.25–5.11)	0.010	4.57(2.01–10.40)	<0.001

All variables are used the baseline data. HbA1c—hemoglobin A1c; MoCA—Montreal Cognitive Assessment.

## Data Availability

The data that support the findings of this study are available from the corresponding author upon reasonable request.

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
