# Peer review of "The Impact of Medication Regimen Adjustment Ratio on Adherence and Glycemic Control in Patients with Type 2 Diabetes and Mild Cognitive Impairment"

_biomedicines, 2024, doi:10.3390/biomedicines12092110_

Round 1
Reviewer 1 Report
Comments and Suggestions for Authors
The paper was to report the effect of the medication regimen adjustment ratio on adherence and glycemic control in patients.
The research experiment design is very rigorous, and the research data collection is also very reliable. However, some issues need to be enhanced before it could be recognized as a research paper.
1. The grammar and style of this paper should be checked totally.
2. Please add the line number on the paper. It was very difficult to note the content of the paper.
3. The section “2.7. Statistical Analyses” is very detailed. However, this section does not introduce the important technique of Logistic regression analysis in detail.
4. Please explain the connection between the statistical method and the results.
For example, in Table 2, what is the data form, and what was the assumption of the statistical technique? Why select the technique for this data set?
5. The expression of the statistical results needs to be checked very carefully. For example, in page 8, “variate analysis confirmed that MoCA score (OR: 0.73, 95% CI:0.62-0.87, P = 0.000)”, that was incorrect. The p-value could not be zero. Please ask for help from a statistician to recheck all statistical tests.
Comments on the Quality of English LanguageExtensive editing of English language required.
Author Response
- Summary
Dear Reviewer:
Thank you for the opportunity to submit a revised draft of our manuscript, “The Impact of Medication Regimen Adjustment Ratio on Adherence and Glycemic Control in Patients with Type 2 Diabetes and Mild Cognitive Impairment” (Manuscript No: 3180394). We deeply appreciate the time and effort you have dedicated to reviewing our work.
We are particularly grateful for the insightful comments and valuable suggestions provided. These comments have been instrumental in guiding our revisions and improving the quality of our paper.
We have carefully reviewed and addressed each of the comments and have made the necessary corrections. The revised sections are marked in red in the manuscript for your convenience. We hope that these revisions meet with your approval.
Thank you once again for your consideration and guidance.
- Point-by-point response to Comments and Suggestions for Authors
Comments 1: The grammar and style of this paper should be checked totally.
Response 1: Thanks for your suggestion. We agree that the manuscript will be greatly improved through language modifications. And we have tried our best to refine and improve the language of the revised manuscript. These changes will not influence the content and framework of the manuscript. Here we did not list the changes but marked them in red in the revised manuscript. We appreciate for Reviewers' warm work earnestly and hope that the correction will meet with approval.
Comments 2: Please add the line number on the paper. It was very difficult to note the content of the paper.
Response 2: Thank you for the helpful suggestion. We have added the line number on the paper.
Comments 3: The section “2.7. Statistical Analyses” is very detailed. However, this section does not introduce the important technique of Logistic regression analysis in detail.
Response 3: We apologize for not describing clearly about Logistic regression analysis. We revised the “2.8. Statistical Analyses” section in the manuscript (lines 165-172) as follows:
“To identify the independent associated variable for non-adherence or worse glycemic control, we used a univariate logistic regression model to identify multiple variables. Then, we applied a multivariate stepwise logistic regression model to exclude confounding factors and identify independent factors significantly associated with medication nonadherence or poor glycemic control. After excluding collinearity, the variables selected for inclusion in the multivariate stepwise model were sex, age, low education, HbA1c%, MoCA, insulin use, and change ratio of medication regimen.”
Comments 4: Please explain the connection between the statistical method and the results.
For example, in Table 2, what is the data form, and what was the assumption of the statistical technique? Why select the technique for this data set?
Response 4: Thank you. We found these comments very helpful. As your suggestion, we detailed the connection between the statistical method and the results as follows:
For Table 1 to Table3, first, the modified Kolmogorov–Smirnov test was used to test the normal distribution of continuous variables,showing that the continuous variables were all non-normally distributed. Second, data are presented as median with interquartile range for non-normally distributed continuous variables and as counts (percentages) for categorical variables. Then, the assumption of the statistical techniques for comparing these three groups is that there are no statistical differences among the three groups. For variables with non-normal distributions among the three groups, the Kruskal–Wallis test was used to evaluate the differences between groups with different cognitive status or different ratios of antidiabetic medication adjustment. And the χ2 test or Fisher’s exact test was used for categorical variables in these groups. P < 0.05 was considered statistically significant. And it indicates a statistical difference among the three groups. Otherwise, it is considered that there is no statistical difference among the three groups.
For Table 4 and Table5, to identify the independent associated variable for non-adherence or worse glycemic control, we used a univariate logistic regression model to identify multiple variables. Then, we applied a multivariate stepwise logistic regression model to exclude confounding factors and identify independent factors significantly associated with medication nonadherence or poor glycemic control. The assumption of the statistical techniques was that the variable was an independent associated factor. If P < 0.05, the hypothesis is considered valid.
Comments 5: The expression of the statistical results needs to be checked very carefully. For example, in page 8, “variate analysis confirmed that MoCA score (OR: 0.73, 95% CI:0.62-0.87, P = 0.000)”, that was incorrect. The p-value could not be zero. Please ask for help from a statistician to recheck all statistical tests.
Response 5: Thank you for your professional advice, and we apologize for this mistake. The description of P=0.000 was indeed incorrect. We have asked the help from a statistician to recheck all the statistical tests. The more accurate description should be P<0.001. We have corrected P=0.000 to P<0.001 throughout the entire manuscript. The revision in the texts (line 236,241,271) have been marked in red. Additionally, Tables 4 and 5 have also been revised in the text (page11,12) as follows:
Table4. Logistic regression analysis of factors associated with medication non-adherence |
|||||||
Variables |
Univariable |
|
Multivariate |
||||
OR (95%CI) |
P |
|
OR (95%CI) |
P |
|||
Male |
0.69 (0.38-1.27) |
0.234 |
|
0.81 (0.39-1.70) |
0.581 |
||
Age |
1.04 (1.01-1.07) |
0.025 |
|
1.03 (0.99-1.08) |
0.120 |
||
Low education |
3.27 (1.19-9.03) |
0.022 |
|
2.13 (0.67-6.83) |
0.202 |
||
HbA1c% |
1.18 (1.03-1.36) |
0.018 |
|
1.04 (0.86-1.25) |
0.710 |
||
MoCA |
0.72 (0.62-0.83) |
<0.001 |
|
0.73 (0.62-0.87) |
<0.001 |
||
Insulin using |
3.29 (1.29-8.41) |
0.013 |
|
1.51 (0.49-4.68) |
0.479 |
||
Adjustment ratio |
|
|
|
|
|
||
0-33.3% |
ref |
|
|
Ref |
|
||
33.4-66.6% |
3.14 (1.10-8.94) |
0.032 |
|
3.22 (1.01-10.24) |
0.048 |
||
66.7-100% |
9.48 (3.68-24.39) |
<0.001 |
|
10.90 (3.63-32.76) |
<0.001 |
||
All variables are used the baseline data. MCI, mild cognitive impairment; HbA1c, hemoglobin A1c; MoCA, Montreal Cognitive Assessment. |
|||||||
Table 5. Logistic regression analysis of factors associated with worse glycemic control |
|||||||
Variables |
Univariable |
|
Multivariate |
||||
OR (95%CI) |
P |
|
OR (95%CI) |
P |
|||
Male |
1.14 (0.63-2.07) |
0.669 |
|
1.32(0.69-2.51) |
0.408 |
||
Age |
1.03 (1.00-1.06) |
0.083 |
|
1.02 (0.99-1.06) |
0.250 |
||
Low education |
1.51 (0.67-3.37) |
0.318 |
|
|
|
||
HbA1c% |
0.87 (0.76-0.99) |
0.039 |
|
0.77 (0.65-0.90) |
0.002 |
||
MoCA |
0.92 (0.81-1.05) |
0.226 |
|
0.92 (0.79-1.06) |
0.262 |
||
Insulin using |
0.89 (0.42-1.89) |
0.759 |
|
|
|
||
Adjustment ratio |
|
|
|
|
|
||
0-33.3% |
ref |
|
|
ref |
|
||
33.4-66.6% |
1.91 (0.86-4.22) |
0.111 |
|
2.70 (1.13-6.41) |
0.025 |
||
66.7-100% |
2.53 (1.25-5.11) |
0.010 |
|
4.57(2.01-10.40) |
<0.001 |
||
All variables are used the baseline data. MCI, mild cognitive impairment; HbA1c, hemoglobin A1c; MoCA, Montreal Cognitive Assessment. |
|||||||
- Response to Comments on the Quality of English Language.
Point 1:Extensive editing of English language required.
Response 1: Thanks for your suggestion. We agree that the manuscript will be greatly improved through language modifications. And we have tried our best to refine and improve the language of the revised manuscript. These changes will not influence the content and framework of the manuscript. Here we did not list the changes but marked them in red in the revised manuscript. We appreciate for Reviewers' warm work earnestly and hope that the correction will meet with approval.
- Additional clarifications
Once again, thank you very much for your comments and suggestions.
Finally, if you have any questions, please feel free to contact us.

Reviewer 2 Report
Comments and Suggestions for Authors
Comments:
The manuscript describes "The Impact of Medication Regimen Adjustment Ratio on Adherence and Glycemic Control in Patients with Type 2 Diabetes and Mild Cognitive Impairment.” Type 2 diabetes (T2DM) and cognitive impairment have a bidirectional relationship. This study investigated the effects of adjusting antidiabetic drug therapy on medication adherence, glycemic control, and cognitive function in T2DM patients with mild cognitive impairment (MCI). 364 consecutive hospitalized T2DM patients were included. During the 3-month follow-up, the HbA1c level in the group with a high rate of antidiabetic drug change was increased, and the MoCA, MMSE, and MARS-5 scores in the high-change group were significantly higher. Therefore, cognitive status should be considered when adjusting medication, and additional guidance may be required for patients with cognitive impairment, but several points need clarification.
Comment:
1. HbA1c values ​​tracked for 3 months; whether the time is too short. It is difficult to objectively analyze the results. The authors should explain clearly.
2. Many abbreviations in the article should be marked.
3. There were three groups based on cognitive status: normal, mild cognitive impairment, and dementia. Dementia patients have greater variation factors, so more relevant conclusions should be written in the discussion. The authors should explain clearly.
4. Why should the total number of patients taking more than 4 anti-diabetic drugs be ranked? The authors should explain clearly.
5. How does the author delineate the change ratio of antidiabetic drugs? 0-33% OR 0-50%
Comments on the Quality of English LanguageMinor editing of English language required.
Author Response
Please see the attachment.
Response to Reviewer 2 Comments
- Summary
Dear Reviewer:
Thank you for the opportunity to submit a revised draft of our manuscript, “The Impact of Medication Regimen Adjustment Ratio on Adherence and Glycemic Control in Patients with Type 2 Diabetes and Mild Cognitive Impairment”, (Manuscript No: 3180394). We deeply appreciate the time and effort you have dedicated to reviewing our work. These comments are all valuable and helpful for improving our article. In this revised version, changes to our manuscript were all marked in red. Thank you once again for your consideration and guidance.
- Point-by-point response to Comments and Suggestions for Authors
Comments 1: HbA1c values tracked for 3 months; whether the time is too short. It is difficult to objectively analyze the results. The authors should explain clearly.
Response 1: Thank you for your valuable suggestions. The reason that we followed up all patients at 3 months after the discharge was that diabetic patients are generally advised to have their HbA1c assessed every 3 months, and for patients with poor blood glucose control, the hypoglycemic regimen might require further adjustment. However, we totally agree that longer follow-up is more beneficial to assess the long-term impact of changes in glycemic regimens on patients, and we will conduct longer follow-ups in future studies.
We added the following text in the Discussion (line340-342): “Second, the relatively short follow-up period may have limited our ability to detect substantial changes in Medication adherence, glycemic control, and MoCA scores between enrollment and follow-up.”
Comments 2: Many abbreviations in the article should be marked.
Response 2: Thank you for the helpful suggestion. We have added the full names of abbreviations in the abstract, main text, figures, and their captions where they first appear. All revisions were marked in red in the text. Additionally, we have included an 'Abbreviations' section at the end of the main text (lines 383-386) as follows:
“Abbreviations
HbA1c: hemoglobin A1c; MCI: mild cognitive impairment; MARS-5: Medication Adherence Report Scale; MMSE: Mini-mental State Examination; MoCA: Montreal Cognitive Assessment; T2DM: Type 2 diabetes mellitus.”
Comments 3: There were three groups based on cognitive status: normal, mild cognitive impairment, and dementia. Dementia patients have greater variation factors, so more relevant conclusions should be written in the discussion. The authors should explain clearly.
Response 3: This is a very meaningful suggestion. Dementia and MCI are two distinct cognitive states with notable differences in memory, daily living skills, social abilities, and other aspects. MCI (Mild Cognitive Impairment) is an intermediate state between normal cognition and dementia. MCI patients experience only mild cognitive impairment, with their daily living abilities remaining largely unaffected. As a result, these patients are more likely to cooperate with follow-up procedures, such as returning to the clinic for face-to-face interviews, completing the medication adherence scale MARS-5, and undergoing cognitive assessments with MoCA and MMSE scales. Additionally, cognition in MCI patients is reversible and represents a critical period for intervention. Previous studies estimate that the prevalence of MCI in patients with T2DM (Type 2 Diabetes Mellitus) is around 45.0% (DOI: 10.1007/s00592-020-01648-9). Meta-analyses have shown that 14%–55% of MCI cases can revert to normal cognition (DOI: 10.1212/WNL.0000000000004826), while the risk of progression from MCI to dementia is 1.53 times higher in T2DM patients compared to non-T2DM patients (DOI: 10.1016/j.arr.2019.100944). Our study focuses on patients with T2DM and MCI, and we conducted follow-ups to explore whether adjustments in antidiabetic medication can improve blood glucose levels and subsequently improve cognitive function.
Dementia patients show significant declines in memory, daily living skills, social abilities, and work capacity, making follow-ups more challenging. This requires more team members, longer follow-up periods, and significant support from family members, who must invest additional time and effort to assist patients throughout the follow-up process.
In this study, patients with type 2 diabetes and dementia were assessed for medication adherence using the MARS-5, and cognition status were assessed with MoCA and MMSE during their hospitalization. The analysis revealed that these patients self-reported high medication adherence. However, this might be due to memory biases in dementia patients. Therefore, assessing medication adherence in dementia patients may require more objective methods or at least two adherence assessment methods for a comprehensive evaluation. This presents a new direction worth exploring for future research. We can further follow up these patients, possibly through prospective studies with larger number of cases, to observe the relationship between medication, glycemic control, and cognition status in T2DM patients with dementia.
Comments 4: Why should the total number of patients taking more than 4 anti-diabetic drugs be ranked? The authors should explain clearly.
Response 4: We sincerely thank the reviewer for carefully reading the manuscript. We did not express this clearly in the manuscript. The patient using more than four antidiabetic drugs was not excluded but was included in the analysis of the 194 T2DM patients with MCI. However, in the subgroup analysis, there was only one patient in the study using more than four types of antidiabetic drugs. As this is the only case, it was categorized into Subgroup 5, but due to the small sample size, statistical analysis could not be performed. We have revised the subgroup classification in Figure 1(Page 6, Line 204), displaying this patient in Subgroup 5. We hope this clears up your confusion.
Revised Figure 1 is shown as follow:
Figure1. Flow chart of the study population.
Comments 5: How does the author delineate the change ratio of antidiabetic drugs? 0-33% OR 0-50%
Response 5: Thank you for valuable comment. We have added the detailed interval range definition range under “2.7. Adjustment Ratio of Medication" in the text (lines 146-156) as follows:
“Changes in the antidiabetic medication regimen at discharge were quantified using the medication adjustment ratio, calculated as follows:
Adjustment ratio = (the number of newly added antidiabetic drug types) / (total number of the antidiabetic drug types at discharge) Í100%.
Ratios were categorized into low (0-33.3%), moderate (33.4-66.6%), and high (66.7-100%).”
Additionally, Supplemental Table 1 presents a subgroup analysis of a single anti-diabetic drug in MCI patients, where the adjustment ratio is categorized into two groups: 0% and 100%. In Supplemental Table 2, which focuses on a subgroup analysis of two or three oral anti-diabetic drugs in MCI patients, the adjustment ratio is divided into two groups: <50% and ≥50%.
The above is the classification of medication change rates in this study. I hope it clarifies your doubts.
- Response to Comments on the Quality of English Language.
Point 1:Minor editing of English language required.
Response 1: Thanks for your suggestion. We have invited a colleague with experience in English Speaker to help refine our article. We hope the revised manuscript will be acceptable to you.
- Additional clarifications
We tried our best to improve the manuscript and made some changes marked in red in revised paper which will not influence the content and framework of the paper. We appreciate for Reviewer’s warm work earnestly, and hope the correction will meet with approval. Once again, thank you very much for your comments and suggestions.
Finally, if you have any questions, please feel free to contact us.

Reviewer 3 Report
Comments and Suggestions for Authors
My specific suggestions for the improvement of the paper:
1. The Introduction mentions the relationship between type 2 diabetes mellitus (T2DM) and mild cognitive impairment (MCI), but it does not adequately specify what is unknown or why this study is necessary. To improve, explicitly highlight the existing gaps in literature concerning medication regimen adjustments and their effects on both glycemic control and cognitive function in patients with T2DM and MCI.
2. While the background provides some context on T2DM and MCI, it lacks focus and relevance. The Introduction could be improved by narrowing down the background information to directly support the hypothesis and objectives of the study. For example, rather than giving a broad overview of diabetes and cognitive impairment, it could focus more specifically on how medication adherence and regimen adjustments impact these conditions. This will make the background more pertinent and concise.
3. The paper lacks a clear and testable hypothesis that guides the research. The objectives are also vaguely stated. To enhance the clarity and direction of the study, the authors should explicitly state a hypothesis based on the identified research gap. Additionally, the objectives should be more precise and directly related to the research questions, making it easier for the reader to understand the aim of the study and the rationale behind the chosen methodology.
4. The Literature Review section cites several studies, but many references are outdated or tangential to the main research focus. To improve, the authors should integrate more recent studies that directly relate to medication adherence, cognitive function, and glycemic control in T2DM patients. Emphasizing recent findings will not only update the literature review but also strengthen the foundation of the study by aligning it with current research trends.
5. The paper uses various terms that may not be familiar to all readers, such as "MARS-5 scale" or "MoCA." To improve readability, the authors should use consistent terminology throughout the paper and avoid excessive jargon. When specific terms or scales are introduced, a brief explanation or definition should be provided to ensure that the text is accessible to a broader audience.
6. The paper briefly outlines the methods for assessing medication adherence, glycemic control, and cognitive status but lacks a comprehensive explanation of how these assessments were conducted. For example, while the MARS-5 scale and the Chinese versions of MoCA and MMSE are mentioned, the process of administering these tests, including the environment and timing, is not described. This lack of detail could affect the reproducibility of the study.
7. Provide a step-by-step description of the methodology, including how participants were prepared for the tests, the specific procedures followed during the assessment, and any measures taken to minimize bias or variability. This would help readers understand how the data was collected and enhance the study's reliability and validity.
8. e study involves inpatients from a single hospital, which could introduce selection bias. The patients’ clinical characteristics and the hospital setting might not be representative of the broader population of T2DM patients with mild cognitive impairment.
9. Clearly define the criteria used for adjusting medication regimens. Include information on any guidelines followed, factors considered by physicians, and how decisions were standardized across participants. This would provide clarity and allow for a better understanding of the intervention’s impact.
10. The study does not adequately address potential confounding variables that could impact the outcomes, such as differences in patients' baseline characteristics, co-morbidities, or socio-economic status, which might influence both medication adherence and glycemic control.
11. The study design appears to be observational without any randomization or blinding, which could introduce bias in the assessment of outcomes. Participants and investigators knowing the group allocations could influence both adherence and reporting of results.
12. The results section does not provide sufficient subgroup analyses, especially for variables that could significantly influence outcomes, such as age, duration of diabetes, and cognitive function levels. These factors might interact differently with medication adherence and glycemic control. Conduct and report subgroup analyses to explore how different patient characteristics might influence the outcomes. This would provide deeper insights into which populations might benefit most or least from certain medication regimen adjustments.
13. The interpretation of the results regarding the association between medication regimen adjustments and outcomes is somewhat vague. The study claims that excessive adjustments might negatively impact outcomes, but it is unclear what constitutes "excessive" adjustment and why these effects might occur. Provide a clearer interpretation of what is meant by "excessive" adjustment and offer a more detailed discussion on the mechanisms by which medication changes might affect adherence, glycemic control, and cognitive function. This would strengthen the conclusions drawn from the study.
Comments on the Quality of English Language1. Moderate English revisions required.
Author Response
Please see the attachment.
- Summary
Dear Reviewer:
We greatly appreciate your professional review of our manuscript, “The Impact of Medication Regimen Adjustment Ratio on Adherence and Glycemic Control in Patients with Type 2 Diabetes and Mild Cognitive Impairment” (Manuscript No: 3180394). Your comments are all valuable and greatly assist in revising and improving our paper, as well as the important guiding significance to our research. We have studied the nice comments carefully and have made extensive corrections and supplemented extra data to make our results convincing which we hope meet with your approval. The revised sections are marked in red in the manuscript.
- Point-by-point response to Comments and Suggestions for Authors
Comments 1: The Introduction mentions the relationship between type 2 diabetes mellitus (T2DM) and mild cognitive impairment (MCI), but it does not adequately specify what is unknown or why this study is necessary. To improve, explicitly highlight the existing gaps in literature concerning medication regimen adjustments and their effects on both glycemic control and cognitive function in patients with T2DM and MCI.
Response 1: We sincerely appreciate the professional comments. We agree that the manuscript will be greatly improved by revising in accordance with the comments. We have added the latest reports on the impact of medication regimen factors on blood glucose control, medication adherence, and cognitive function in patients with type 2 diabetes, as well as existing gaps, to more clearly indicate the purpose and significance of this study. We have re-written the “1. Introduction” section according to the Reviewer’s suggestion in the text (lines 64-76) as follows:
“Recent studies have primarily focused on the impact of medication regimen complexity on adherence and glycemic control [16-18]. Additionally, there is limited research on medication adherence in diabetes patients with cognitive impairment. Some studies have found no association between dementia and adherence to antidiabetic medications [19], while others have highlighted how negative beliefs about medications and regimen-related distress affect adherence and glycemic control in patients with diabetes and mild cognitive impairment (MCI) [20]. Currently, there is a lack of research on the impact of medication regimen adjustments on adherence, glycemic control, and cognitive status in T2DM patients with cognitive impairment. Understanding the relationship between medication regimen adjustments and adherence, and glycemic control may help in developing future interventions to improve treatment outcomes. This study aims to explore the impact of changes in anti-diabetic medication regimens on medication adherence, glycemic control, and cognitive status in T2DM patients with MCI.”
Comments 2: While the background provides some context on T2DM and MCI, it lacks focus and relevance. The Introduction could be improved by narrowing down the background information to directly support the hypothesis and objectives of the study. For example, rather than giving a broad overview of diabetes and cognitive impairment, it could focus more specifically on how medication adherence and regimen adjustments impact these conditions. This will make the background more pertinent and concise.
Response 2: Thank you for your professional suggestions. We have made the following revisions to the introduction. Simplified the sections on T2DM and MCI, while adding information on how medication regimen factors influence medication adherence and glycemic control. Additionally, we have included a discussion on medication adherence in patients with T2DM and mild cognitive impairment. We hope these changes will help narrow the background information and better focus the study. Revisions in the text (lines 64-76) are detailed as follows:
“Recent studies have primarily focused on the impact of medication regimen complexity on adherence and glycemic control [16-18]. Additionally, there is limited research on medication adherence in diabetes patients with cognitive impairment. Some studies have found no association between dementia and adherence to antidiabetic medications [19], while others have highlighted how negative beliefs about medications and regimen-related distress affect adherence and glycemic control in patients with diabetes and mild cognitive impairment (MCI) [20]. Currently, there is a lack of research on the impact of medication regimen adjustments on adherence, glycemic control, and cognitive status in T2DM patients with cognitive impairment. Understanding the relationship between medication regimen adjustments and adherence, and glycemic control may help in developing future interventions to improve treatment outcomes. This study aims to explore the impact of changes in anti-diabetic medication regimens on medication adherence, glycemic control, and cognitive status in T2DM patients with MCI.”
Comments 3: The paper lacks a clear and testable hypothesis that guides the research. The objectives are also vaguely stated. To enhance the clarity and direction of the study, the authors should explicitly state a hypothesis based on the identified research gap. Additionally, the objectives should be more precise and directly related to the research questions, making it easier for the reader to understand the aim of the study and the rationale behind the chosen methodology.
Response 3: Thank you for the pertinent suggestion. We apologize for not clearly stating the hypothesis and objectives in the Introduction. By reviewing additional literature, we have provided a clearer introduction to the background, refined the hypothesis, and more clearly articulated the research objectives.
We revised the following part in the”1. Introduction” section (lines 70-76):
“Currently, there is a lack of research on the impact of medication regimen adjustments on adherence, glycemic control, and cognitive status in T2DM patients with cognitive impairment. Understanding the relationship between medication regimen adjustments and adherence, and glycemic control may help in developing future interventions to improve treatment outcomes. This study aims to explore the impact of changes in anti-diabetic medication regimens on medication adherence, glycemic control, and cognitive status in T2DM patients with MCI.”
Comments 4: The Literature Review section cites several studies, but many references are outdated or tangential to the main research focus. To improve, the authors should integrate more recent studies that directly relate to medication adherence, cognitive function, and glycemic control in T2DM patients. Emphasizing recent findings will not only update the literature review but also strengthen the foundation of the study by aligning it with current research trends.
Response 4: Thank you for your insightful comments. We strongly agree that proper citation of references is indeed very important. Incorporating new references can not only clarify the current research trends, but also reinforces the foundation of this research. We updated the citations according to the Reviewer’s suggestion in the "Introduction" section are as follows:
“8. American Diabetes Association. Standards of Medical Care in Diabetes-2021 Abridged for Primary Care Providers. Clin Diabetes. 2021;39(1):14-43.
- Evans M, Engberg S, Faurby M, Fernandes JDDR, Hudson P, Polonsky W. Adherence to and persistence with antidiabetic medications and associations with clinical and economic outcomes in people with type 2 diabetes mellitus: A systematic literature review. Diabetes Obes Metab. 2022;24(3):377-390.
- Shalaeva EV, Bano A, Kasimov U, Janabaev B, Laimer M, Saner H. Impact of Persistent Medication Adherence and Compliance with Lifestyle Recommendations on Major Cardiovascular Events and One-Year Mortality in Patients with Type 2 Diabetes and Advanced Stages of Atherosclerosis: Results From a Prospective Cohort Study. Glob Heart. 2023;18(1):61. Published 2023 Nov 1. doi:10.5334/gh.1273
- Ab Rahman N, Lim MT, Thevendran S, Ahmad Hamdi N, Sivasampu S. Medication Regimen Complexity and Medication Burden Among Patients With Type 2 Diabetes Mellitus: A Retrospective Analysis. Front Pharmacol. 2022; 13:808190.
- 111 B. Ayele AA, Tegegn HG, Ayele TA, Ayalew MB. Medication regimen complexity and its impact on medication adherence and glycemic control among patients with type 2 diabetes mellitus in an Ethiopian general hospital. BMJ Open Diabetes Res Care. 2019;7(1): e000685.
- 111 C. Russell AM, Opsasnick L, Yoon E, Bailey SC, O'Brien M, Wolf MS. Association between medication regimen complexity and glycemic control among patients with type 2 diabetes. J Am Pharm Assoc (2003). 2023;63(3):769-777.
- 111 Mirghani H, Aljohani S, Albalawi A. Dementia and Adherence to Anti-Diabetic Medications: A Meta-Analysis. Cureus. 2021;13(4): e14611. Published 2021 Apr 21.
- Rovner BW, Casten RJ. Health Beliefs and Medication Adherence in Black Patients with Diabetes and Mild Cognitive Impairment. Am J Geriatr Psychiatry. 2018;26(7):812-816.”
Comments 5: The paper uses various terms that may not be familiar to all readers, such as "MARS-5 scale" or "MoCA." To improve readability, the authors should use consistent terminology throughout the paper and avoid excessive jargon. When specific terms or scales are introduced, a brief explanation or definition should be provided to ensure that the text is accessible to a broader audience.
Response 5: Thank you for the helpful suggestion. We have added the full names of abbreviations in the abstract, main text, figures, and their captions where they first appear. All revisions were marked in red in the text. Additionally, we have included an 'Abbreviations' section at the end of the main text (lines 383-386) as follows:
“Abbreviations
HbA1c: hemoglobin A1c; MCI: mild cognitive impairment; MARS-5: Medication Adherence Report Scale; MMSE: Mini-mental State Examination; MoCA: Montreal Cognitive Assessment; T2DM: Type 2 diabetes mellitus.”
Comments 6: The paper briefly outlines the methods for assessing medication adherence, glycemic control, and cognitive status but lacks a comprehensive explanation of how these assessments were conducted. For example, while the MARS-5 scale and the Chinese versions of MoCA and MMSE are mentioned, the process of administering these tests, including the environment and timing, is not described. This lack of detail could affect the reproducibility of the study.
Response 6: Thank you for the valuable comment. We have added a new section, “2.6. Measures,” to detail how the assessments of MARS-5, MoCA, and MMSE were conducted. Please refer to "Response 7" for detailed revisions in the text.
Comments 7: Provide a step-by-step description of the methodology, including how participants were prepared for the tests, the specific procedures followed during the assessment, and any measures taken to minimize bias or variability. This would help readers understand how the data was collected and enhance the study's reliability and validity.
Response 7: Thank you. In response to questions 6 and 7, we have made comprehensive revisions to”2. Materials and methods’’. We added a new section” 2.6 Measures”, which includes a detailed description of the preparations made before patient testing, including the setting, environment, and the atmosphere of communication with patients. Additionally, we have provided a more detailed description of the testing procedures.
The revisions in the text (lines 124-145) were as follows:
“ 2.6. Measures
Medication adherence and cognition status were assessed through interviews at admission and a 3-month follow-up.
MoCA and MMSE Assessment:
1.Conducted in a dedicated, quiet room with necessary materials provided,and no clocks or calendars are present.
2.Each participant takes the test face-to-face.
3.A 5-minute calming conversation precedes the assessment to help the participant relax.
4.Each test item is attempted only once, with neutral feedback.
5.The assessment lasts about 10 minutes with uniform instructions and adherence to "Scoring Criteria."
MARS-5 Assessment:
- Assessed medication adherence with 5 questions (e.g., “I forget to take my anti-diabetic drugs”; “I alter the dose of my antidiabetic drugs”; “I stop taking my antidiabetic drugs for a while/sometimes”; “I decide to skip one of my antidiabetic drugs dosages”; “I use my antidiabetic drugs less than is prescribed”, using a 5-level response format (1—always, 2—often, 3—sometimes, 4—rarely, and 5—never).
- Scores range from 5 to 25, with higher scores indicating better adherence.
All assessments were performed by clinicians trained for these tasks.”
Comments 8: the study involves inpatients from a single hospital, which could introduce selection bias. The patients’ clinical characteristics and the hospital setting might not be representative of the broader population of T2DM patients with mild cognitive impairment.
Response 8: We agree that this is a potential limitation of the study. We have added this as a limitation in “Discussion” in the text (lines 345-348) as follow:
“Finally, this is a single-center study with a single source of patients, and the patients' clinical characteristics and hospital environment may not be representative of the broader population of T2DM patients with mild cognitive impairment.”
Comments 9: Clearly define the criteria used for adjusting medication regimens. Include information on any guidelines followed, factors considered by physicians, and how decisions were standardized across participants. This would provide clarity and allow for a better understanding of the intervention’s impact.
Response 9: Thank you for the helpful suggestion. We have added information on Materials and methods section (lines 153-156) as follows:
“The adjustment of antidiabetic medications for all inpatients was a comprehensive evaluation performed by attending physicians with over 10 years of clinical experience, based on the latest diabetes treatment guidelines [35-37], clinical experience, and the patient's blood glucose and HbA1c levels during hospitalization.”
Comments 10: The study does not adequately address potential confounding variables that could impact the outcomes, such as differences in patients' baseline characteristics, co-morbidities, or socio-economic status, which might influence both medication adherence and glycemic control.
Response 10: Thank you for your insightful suggestions, which we fully agree with. We have added data on baseline characteristics and comorbidities of the 194 T2DM patients with MCI, including gender, age, diabetes duration, BMI, smoking and alcohol history, hyperlipidemia, hypertension, and diabetes complications, and performed statistical analysis. These data have been included in the revised Table 2. The results indicate no significant differences were observed among the three groups for these variables, suggesting that these factors may be not major influences on medication adherence and glucose control in this study. The revised Table 2 (page7-8) is as follow:
Table2. Characteristics of the MCI group categorized by adjustment ratio of anti-diabetic drugs (N=194) |
||||
Characteristics
|
Adjustment ratio of anti-diabetic drugs |
P |
||
0-33.3% (n=50) |
33.4-66.6% (n=50) |
66.7-100% (n=94) |
||
Baseline |
|
|
|
|
Age (years) |
58.0 (52.0 ,62.0) |
59.5 (52,62.3) |
58.0 (52.0,62.0) |
0.866 |
Gender |
|
|
|
|
Male, n (%) |
30 (60) |
38 (76) |
60 (63.8) |
0.199 |
Female, n (%) |
20 (40) |
12 (24) |
34 (36.2) |
|
Diabetes duration (years) |
10.0 (4.0,13.3) |
8.0 (5.0,15.3) |
23.8 (21.9,25.3) |
0.181 |
BMI (kg/m2) |
23.3 (22.3,24.9) |
23.8 (21.9,25.3) |
23.2 (22.2,25.0) |
0.934 |
Hypoglycemic event, n (%) |
3 (6.0) |
3 (6.0) |
2 (2.1) |
0.401 |
Hyperlipidemia, n (%) |
32() |
33() |
73() |
0.169 |
Hypertension, n (%) |
22 (44.0) |
24 (48.0) |
46 (48.9) |
0.849 |
Smoker, n (%) |
7 (14.0) |
9 (18.0) |
10 (10.6) |
0.462 |
Drinker, n (%) |
6 (12.2) |
7 (14.0) |
9 (9.7) |
0.727 |
Diabetes Complications, n (%) |
|
|
|
|
|
|
|
|
|
Diabetic Foot |
2 (4.0) |
3 (6.0) |
2 (2.1) |
0.488 |
Diabetic Retinopathy |
11 (22.0) |
15 (30..0) |
30 (31.9) |
0.448 |
Diabetic Nephropathies |
16 (32.0) |
17 (34.0) |
38 (40.4) |
0.550 |
Diabetic Peripheral Neuropathies |
32 (64.0) |
31 (62.0) |
64 (68.1) |
0.741 |
Diabetic Peripheral vascular disease |
38 (76) |
37 (74) |
79 (84) |
0.289 |
Data are expressed as median (quartile) or n (%). MCI, mild cognitive impairment; HbA1c, hemoglobin A1c; MoCA, Montreal Cognitive Assessment; MMSE, Mini-mental State Examination; MARS-5, the Medication Adherence Report Scale-5; ΔHbA1c, HbA1c 3-month - HbA1c baseline; ΔMoCA, MoCA3-month -MoCA baseline; ΔMMSE, MMSE3-month -MMSE baseline.
|
Although, as the reviewer pointed out, socio-economic status may influence patient adherence and glucose control, we regret that our data does not include this information. Economic conditions are not directly related to the treatment of hospitalized patients and involve personal privacy beyond medical care, making it challenging for healthcare professionals to obtain such details. We therefore apologize for and regret this limitation.
Comments 11: The study design appears to be observational without any randomization or blinding, which could introduce bias in the assessment of outcomes. Participants and investigators knowing the group allocations could influence both adherence and reporting of results.
Response 11: Thank you for your suggestion. As you pointed out, our study is an observational cross-sectional study, which is also mentioned in the “Abstract Background” section. Additionally, in the ”2. Materials and Methods “section we described that the study involved a 3-month follow-up of discharged patients. Our study did not include randomized grouping or blinding., and no interventions beyond routine medical care were implemented. We collected clinical data, as well as MARS-5, MoCA, and MMSE scores, at discharge (i.e., baseline) and 3-month follow-up. In the statistical analysis, we assessed whether medication adjustments had an impact on these aspects by comparing changes in patients' medication adherence, blood glucose control, and cognitive status at the 3-month follow-up with baseline levels. We hope this clarifies your concerns.
Comments 12: The results section does not provide sufficient subgroup analyses, especially for variables that could significantly influence outcomes, such as age, duration of diabetes, and cognitive function levels. These factors might interact differently with medication adherence and glycemic control. Conduct and report subgroup analyses to explore how different patient characteristics might influence the outcomes. This would provide deeper insights into which populations might benefit most or least from certain medication regimen adjustments.
Response 12: Yes, we agree with the Reviewer’s suggestion that variables such as age, duration of diabetes, and cognitive function levels may interact differently with medication adherence and glycemic control. We recognize the importance of incorporating these variables into our subgroup analyses. Therefore, in each subgroup analysis, we not only included cognitive levels but also added variables such as age and duration of diabetes. It is noteworthy that there were no statistically significant differences in age and duration of diabetes between groups in each subgroup analysis. We have added related content in the revised Table 3, Supplemental table1 and Supplemental table2 in the “3. Results” ( Page 9,10; and Supplemental table) section as follows:
Table3. Subgroups analyze of the MCI patients categorized by anti-diabetic medication regimen |
|||||||
Subgroup
|
Characteristics
|
Adjustment ratio of anti-diabetic drugs |
P |
||||
0-33.3% |
33.4-66.6% |
66.7-100% |
|
||||
1 |
Baseline |
|
|
|
|
||
|
Age (years) |
60.0 (57.0,62.0) |
60.0 (55.0,65.0) |
60.0 (52.0,64.8) |
0.917 |
||
|
Diabetes duration (years) |
10.0 (5.0,13.0) |
14.0 (5.0,20.0) |
7.0 (2.0,16.5) |
0.383 |
||
|
MoCA |
22.0 (20.0,24.0) |
22.0 (19.0,24.0) |
21.0 (20.0,22.8) |
0.932 |
||
|
MMSE |
26.0 (25.0,28.0) |
26.0 (25.0,28.0) |
26 (25.0,28.0) |
0.906 |
||
2 |
Baseline |
|
|
|
|
||
|
N (%) |
13 (19.1) |
17 (25.0) |
38 (55.9) |
|
||
|
Age (years) |
59.0 (51.5,62.5) |
60.0(52.0,62.5) |
55.5 (50.0,61.0) |
0.627 |
||
|
Diabetes duration (years) |
13.0 (2.5, 21.0) |
6.0 (5.0,10.0) |
5.5 (1.8,10) |
0.187 |
||
|
MoCA |
23.0 (20.0,25.0) |
23.0 (21.0,23.5) |
22.5 (20.0,25.0) |
0.785 |
||
|
MMSE |
28.0 (25.0,29.0) |
28.0 (25.52,9.5) |
27.5 (25.0,29.0) |
0.827 |
||
Data are expressed as median (quartile) or n (%). MCI, mild cognitive impairment; Subgroup1, Insulin/glucagon-like peptide-1 receptor agonists combined with an oral anti-diabetic drug; Subgroup2, Insulin/glucagon-like peptide-1 receptor agonists combined with two or three oral anti-diabetic drugs; HbA1c, hemoglobin A1c; MoCA, Montreal Cognitive Assessment; MMSE, Mini-mental State Examination; MARS-5, the Medication Adherence Report Scale-5; ΔHbA1c, HbA1c 3-month - HbA1c baseline; ΔMoCA, MoCA3-month -MoCA baseline; ΔMMSE, MMSE3-month -MMSE baseline. |
|||||||
Supplemental table1. Subgroup analysis of a single anti-diabetic drug in MCI patients (N=35) |
||||
Characteristics |
Adjustment ratio of anti-diabetic drugs |
P |
||
0% |
|
100% |
||
Baseline |
|
|
|
|
Age (years) |
54.0 (49.5,58.5) |
|
58.5 (52.7,62.5) |
0.188 |
Diabetes duration (years) |
10.0 (7.5,12.5) |
|
7.5 (0.0,15.5) |
0.321 |
MoCA |
23.0 (22.0,24.5) |
|
22.0 (20.0,23.0) |
0.159 |
MMSE |
28.0 (26.0,29.0) |
|
26.0 (25.0,28.0) |
0.038 |
Data are expressed as median (quartile) or n (%). MCI, mild cognitive impairment; HbA1c, hemoglobin A1c; MoCA, Montreal Cognitive Assessment; MMSE, Mini-mental State Examination; MARS-5, the Medication Adherence Report Scale-5;ΔHbA1c, HbA1c 3-month - HbA1c baseline; ΔMoCA, MoCA3-month -MoCA baseline; ΔMMSE, MMSE3-month -MMSE baseline. |
Supplemental table2. Subgroup analysis of two or three oral anti-diabetic drugs in MCI patients (N=24) |
||||
Characteristics |
Adjustment ratio of anti-diabetic drugs |
P |
||
<50% |
|
≥50% |
|
|
Baseline |
|
|
|
|
Age (years) |
62.0 (55.0,66.5) |
|
53.0 (51.0,62.00 |
0.155 |
Diabetes duration (years) |
6.0 (3.0,9.0) |
|
7.0 (3.0, 10.0) |
0.560 |
MoCA |
21.3±2.1 |
|
22.1±1.9 |
0.349 |
MMSE |
26.9±1.7 |
|
27.5±1.7 |
0.454 |
Data are expressed as median (quartile) or n (%). MCI, mild cognitive impairment; HbA1c, hemoglobin A1c; MoCA, Montreal Cognitive Assessment; MMSE, Mini-mental State Examination; MARS-5, the Medication Adherence Report Scale-5; ΔHbA1c, HbA1c 3-month - HbA1c baseline; ΔMoCA, MoCA3-month -MoCA baseline; ΔMMSE, MMSE3-month -MMSE baseline. |
Comments 13: The interpretation of the results regarding the association between medication regimen adjustments and outcomes is somewhat vague. The study claims that excessive adjustments might negatively impact outcomes, but it is unclear what constitutes "excessive" adjustment and why these effects might occur. Provide a clearer interpretation of what is meant by "excessive" adjustment and offer a more detailed discussion on the mechanisms by which medication changes might affect adherence, glycemic control, and cognitive function. This would strengthen the conclusions drawn from the study.
Response 13: Your suggestion really means a lot to us.
- We agree that the interpretation of the results regarding the association between medication regimen adjustments and outcomes was not sufficiently clear or precise with the "excessive adjustment". We agree that the interpretation of the results regarding the relationship between medication regimen adjustments and outcomes lacked clarity and precision, particularly with regarding to the issue of "excessive adjustment." So, we have revised "excessive adjustment" to "high ratio of medication adjustments" marked in red in the text. Additionally, we have refined range of interval for the ratio of medication adjustment and clarified the classifications of low, medium, and high in the manuscript (lines 155-156) as follows:
“Ratios were categorized into low (0-33.3%), moderate (33.4-66.6%), and high (66.7-100%).”
- As you suggested, we have added more discussion about the mechanisms by which medication adjustments may affect adherence, glycemic control, and cognitive function in the text (lines 335-341) as follows:
“Changes in medication regimens may lead to increased confusion or difficulty in adhering to the prescribed treatments for patients with T2DM and MCI. While poor medication adherence itself leads to worse glycemic control. Previous studies have reported that long-term diabetes and poor blood glucose control (including hyperglycemia, hypoglycemia, and glycemic fluctuations) can lead to neuronal damage and inflammation-related glial activation by disrupting the blood-brain barrier and altering brain metabolism, resulting in progressive neuropathy and ultimately lead to cognitive impairment [45,46].”
- Response to Comments on the Quality of English Language.
Point 1:Moderate English revisions required.
Response 1: Thanks for your suggestion. We agree that the manuscript will be greatly improved through English revisions. And we have tried our best to refine and improve the language of the revised manuscript. These changes will not influence the content and framework of the manuscript. Here we did not list the changes but marked them in red in the revised manuscript. We appreciate for Reviewers' warm work earnestly and hope that the correction will meet with approval.
- Additional clarifications
Once again, thank you very much for your comments and suggestions.
Finally, if you have any questions, please feel free to contact us.

Round 2
Reviewer 1 Report
Comments and Suggestions for Authors
All problems are replied to adequately.
Comments on the Quality of English LanguageMinor editing of English language required.
Reviewer 2 Report
Comments and Suggestions for Authors
Accepted
Comments on the Quality of English LanguageMinor editing of English language required.
Reviewer 3 Report
Comments and Suggestions for Authors
1. Queries are responsibly and reasonably answered including justifications of limitations of the study.
2. Please improve more the clarity and flow of sentences. Remove unnecessary words that slow down the reading experience of the audience.
Comments on the Quality of English Language1. Minor English corrections.